# Nanopore-Level Wood-Water Interactions—A Molecular Simulation Study

**Jingbo Shi [1],* and Stavros Avramidis [2]**

[1]  Department of Wood Science and Engineering, Nanjing Forestry University, 159 Longpan, Nanjing 210037, China
[2]  Department of Wood Science, The University of British Columbia, 2424 Main Mall, Vancouver, BC V6T 1Z4, Canada; stavros.avramidis@ubc.ca
*  Correspondence: shijb@njfu.edu.cn

**Abstract:** The nanoscale wood-water interaction strength, accessible sorption sites, and cell wall pore sizes are important factors that drive water sorption and the hysteresis phenomenon in wood. In this work, these factors were quantitatively studied using molecular simulations based on a cell wall pore model, previously developed by the authors. Specifically, the wall-water interaction strength, the sorption sites network including their number, interaction range, strength, and spatial distributions were set at a series of theoretical values as simulation input parameters. The results revealed that most of the investigated parameters significantly affected both sorption isotherms and hysteresis. Water monolayers and clusters were observed on the simulated pore surface when the wood-water interaction and sorption site strength were set at unrealistically high values. Furthermore, multiple linear regression models suggested that wood-water interaction and sorption site parameters were coupled in determining sorption isotherms, but not in determining hysteresis.

**Keywords:** cell wall pore size; molecular simulations; water sorption; hysteresis; sorption sites; wood-water interaction strength

## 1. Introduction

Owing to bound water's pronounced influence on various physical and mechanical properties of wood and engineered wood products, great efforts have been devoted to study the wood-water fundamentals. A common, but effective way to explore this relationship is to study water sorption isotherms and the associated hysteresis phenomenon. The three major components of wood, namely, cellulose, hemicellulose, and lignin, are hydrophilic, albeit to very different extents with lignin being almost hydrophobic, and the corresponding sorption isotherm displays a sigmoid shape (Type II according to the International Union of Pure and Applied Chemistry (IUPAC) [1]), that shows hysteresis in its entire range.

For almost a century [2], sorption mechanisms and the origin of hysteresis have been extensively studied, however, there are still unanswered questions and some uncertainty [3]. Water layering and clustering are two proposed sorption mechanisms at low to medium relative humidity (*H*) regions [4–7], whereas capillary condensation is argued to take place at high or very high *H* (>98%) regions [8–12]. Corresponding to these three mechanisms, nano-level wood-water interactions, accessible sorption sites (mainly hydroxyl groups) and cell wall pore size are three major factors that drive the sorption process in wood. The roles of accessibility and quantity of sorption sites have been reassessed in some recent studies [13–16].

The studies on the origin of sorption hysteresis in wood (more specifically, reproducible hysteresis in [17]) are quite diverse. These include, but not limited to the roles of accessible sorption sites during sorption and their possible coupling with sorption-induced swelling [18–21], the rigidity or softening of wood cell walls [22–24], and the metastable states of adsorbed and desorbed water associated with the cell wall nanopore filling [25].

Experimentally, the effects of the wood-water interactions, accessible sorption sites and cell wall pore size on sorption and hysteresis are often investigated by modifying wood using various methods, e.g., thermal or chemical, and then relating the hygroscopic property change to changes of the number of accessible hydroxyl groups, wood-water interaction strength (WWIS) and cell wall pore size distributions [26–33]. However, it is difficult to quantitatively evaluate the aforementioned factors and possible couplings among them. In [25], the effect of cell wall pore size on sorption and hysteresis was explored using molecular simulation methods based on a nanoscale cell wall pore model (details can be found in the following Simulation Model and Method section), and promising results were reported. Comparing to the purely theoretical and experimental approach, the molecular simulation approach has the advantage of conducting systematic and quantitative investigations at the nanoscale using a model that is close to the real system. Molecular simulations of wood cell walls at the atomic level is still challenging due to the implicit cell wall structures, and is very time-consuming which limits its usage in parametric investigations [21].

In this work, the effects of sorption sites and wood-water interactions, their potential coupling, and possible interrelations with cell wall pore sizes were explored by molecular simulations based on a cell wall pore model. By exploring the extreme value scenarios of the investigated factors in the simulations, more insights may be gained into the formation of water monolayers and clusters, and therefore the overall water sorption mechanism in wood.

## 2. Simulation Model and Method

The simulation model was taken from our previous publication [25]. The wood cell wall-water system was modeled as a collection of infinitely long cylindrical nanopores of different sizes $d_j$ with water confined in them (Figure 1a). These pores were independent of each other and had full access to the environment. The independence assumption was based on the systematic investigation of experimental water sorption hysteresis patterns [17], and was further supported by a modified Preisach model [34]. The simulated pores were assumed rigid, and the actual swelling of the cell wall and possible generation of new pores during sorption was addressed by assumed cell wall pore size distributions ($\psi_i$) evolving with $H$ (Figure 1c, details can be found in [35]). Consequently, pore walls were modeled as rigid ones composed of equally distributed Lennard-Jones (LJ) [36] carbon and oxygen atoms (Figure 2a).

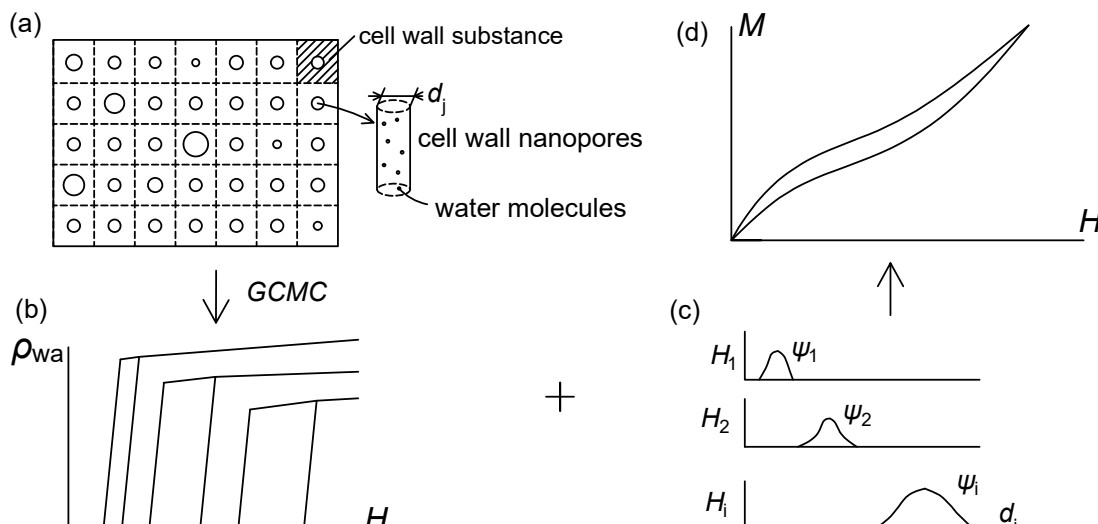

**Figure 1.** Illustration of the wood cell wall nanopore model and modelled sorption isotherms. (**a**) the cell wall substance and nanopores with an initial pore size distribution $\psi_1$; (**b**) simulated sorption isotherms for different pore sizes $d_j$; $\rho_{wa}$ represents the adsorbed water density (**c**) cell wall pore size distributions $\psi_i$ evolving with humidity; (**d**) the eventual wood water sorption isotherm simulated.

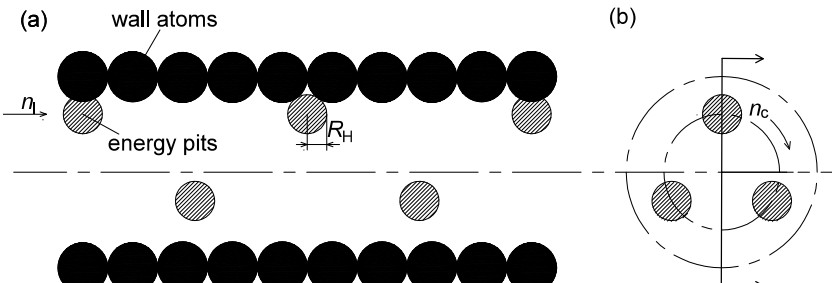

**Figure 2.** Schematic of a cylindrical pore showing energy pits (sorption sites) distributions: (**a**) front view; (**b**) side view.

Two default pore walls, $PW_1$ and $PW_2$ were defined as representatives associated with amorphous holocellulose and lignin, respectively. The accessible sorption sites were modeled as negative energy pits attached to walls (Figure 2), and water was represented by the extended simple point charge (SPC/E) model [37]. Grand canonical Monte Carlo (GCMC) [38] was used to calculate the quantity of adsorbed water molecules at a series of given $H$ under isothermal conditions for each fixed $d_j$. Figure 1b illustrates the simulated sorption isotherms for a series of $d_j$. The final sorption isotherm of wood (Figure 1d) could be obtained by superposition of the simulated sorption isotherms in Figure 1b based on $\psi_i$ at a series of $H_i$ values (Figure 1c). Details can be found in [35].

The interaction energy ($u_{ww}^{ij}$) of any two water molecules $i$ and $j$ was calculated as the sum of the LJ potential and the Coulomb potential [39]; the interaction energy of a water molecule with the pore wall ($U(s_1)$) was calculated by integrating LJ potential over the wall atoms [40]. The LJ parameters ($\varepsilon_{sf}$ and $\sigma_{sf}$, representing the well depth at the minimum interaction energy and the separation where the energy equals 0) of $PW_1$ and $PW_2$-water interactions were taken from [41,42]. Due to the limited computational resources, only one representative cell wall pore diameter of 0.95 nm (an approximate average pore diameter) and one temperature level of 25 °C were selected for the simulations.

WWIS's influence on sorption and hysteresis was investigated by simulating six additional pore walls ($PW_3$–$PW_8$). Varied strengths were set within possible ranges in practice, and at unrealistic extreme values to analyze theoretical trends. $PW_3$ were comprised of pure oxygen atoms, and $PW_4$, pure carbon atoms. WWIS of $PW_5$ to $PW_7$ was enhanced based on default $PW_1$, whereas WWIS of $PW_8$ was decreased based on another default $PW_2$. Table 1 presents the LJ potential parameters of the studied eight wall types. To avoid complications from sorption sites, the number of energy pits on the additional walls was set to 0.

**Table 1.** Summary of LJ potential parameters of eight types of pore walls and water.

| Interaction | LJ Potential Parameters | |
| --- | --- | --- |
| | $\varepsilon_{sf}/k_B$ (K) | $\sigma_{sf}$ (Å) |
| $PW_1$-water | 58.29 | 3.281 |
| $PW_2$-water | 37.60 | 3.190 |
| $PW_3$-water | 78.23 | 3.166 |
| $PW_4$-water | 28 | 3.4 |
| $PW_5$-water | 87.435 | 3.281 |
| $PW_6$-water | 174.87 | 3.281 |
| $PW_7$-water | 291.45 | 3.281 |
| $PW_8$-water | 7.52 | 3.190 |

The influence of accessible sites on sorption was analyzed by controlling the number, strength, interaction range, and spatial distributions of the energy pits on $PW_1$ and $PW_2$. The number of sorption sites $n_H$ was controlled by $n_c$ and $n_l$ ($n_H = n_c \times n_l$), namely, the number of energy pits in the lateral and longitudinal direction of the simulated pore,

respectively (Figure 2). Since the perimeter of the simulated 0.95 nm pore is limited, $n_c$ was kept to be a constant of 3, but values of $n_l$ were adjusted. The strength of the pits ($\varepsilon_H$) was set to be 4.5 kcal mol$^{-1}$ (normal values), 0.5 kcal mol$^{-1}$ (extreme low values), and 22.5 kcal mol$^{-1}$ (extreme high values). The interaction range of the energy pits was evaluated by the pit radius $R_H$ (Figure 2) that were set at 0.317 nm (normal values), 0.158 nm, and 0.475 nm. The spatial distributions of the energy pit were set "even, random, clustered, and helix" (Figure 3). The helix distributions were inspired by recent findings of two- and threefold helix xylan configurations when hydrogen bonding with cellulose fibrils or lignin [43–45]. Table 2 summarizes the assessed settings of energy pits on default $PW_1$ and $PW_2$ cylindrical pores.

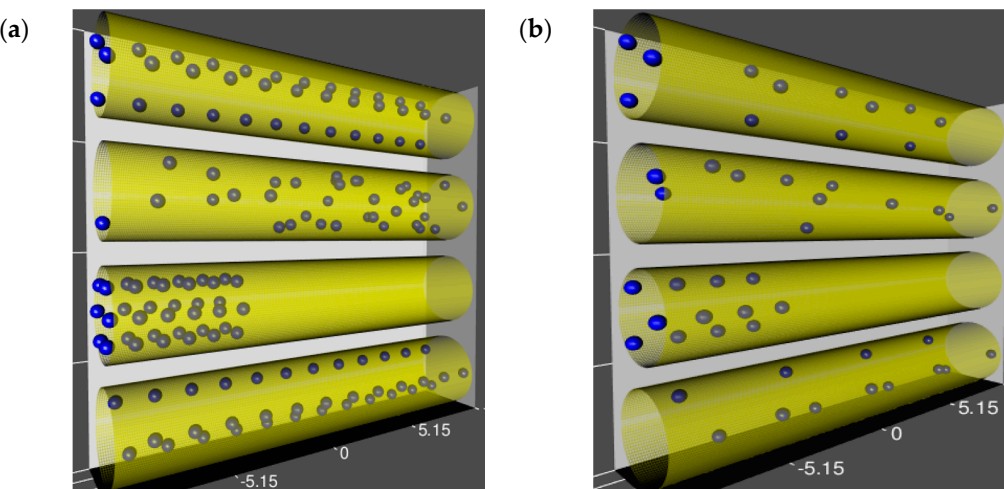

**Figure 3.** Simulated cell wall pores with blues spheres showing the position of energy pits with spatial distributions from the top to the bottom: even, random, clustered, helix. (**a**) $PW_1$ pores; (**b**) $PW_2$ pores.

**Table 2.** Summary of energy pit settings on simulated cylindrical pore walls.

| Wall Type | $n_c$ | $n_l$ | $\varepsilon_H$ (kcal mol$^{-1}$) | $R_H$ (nm) | Spatial Distribution |
|---|---|---|---|---|---|
| $PW_1$ | 3 | 12 | 4.5 | 0.317 | even |
|  | 3 | 16 | 0.5 | 0.158 | random |
|  | 3 | 8 | 22.5 | 0.475 | clustered |
|  |  |  |  |  | helix |
| $PW_2$ | 3 | 4 | 4.5 | 0.317 | even |
|  | 3 | 6 | 0.5 | 0.158 | random |
|  | 3 | 2 | 22.5 | 0.475 | clustered |
|  |  |  |  |  | helix |

## 3. Results and Discussion

### 3.1. The Differences between the Shapes of Simulated and Experimental Sorption Isotherms

Figure 4a,b present the simulated water sorption isotherms for the studied eight types of wood cell wall pores at a fixed diameter of 0.95 nm. The relative density of adsorbed water ($\rho_{wa}$), calculated as the ratio of the density of simulated adsorbed and bulk liquid water at the same temperature and ambient pressure, was used to represent the quantity of adsorbed water molecules. $\rho_{wa}$ could be further connected to $M$ of simulated wood samples by taking into account the mass and chemical composition of simulated pore walls [35]. The shapes of the simulated sorption isotherms were more like those from non-swelling sorption systems, e.g., carbon-water [46,47], carbon nanotube-water [48], carbon-$N_2$/Ar [49], activated carbon-water [50], zeolites-water [51] systems, etc., rather than the ones from the experimental wood water sorption isotherm [23]. This was because

only sorption isotherms for one cell wall pore size, 0.95 nm, were presented here. In the "Simulation Model and Method" section, it was pointed that since the cell wall pores were assumed rigid in the model, the actual swelling of cell walls during sorption had to be addressed by assumed $\psi_i$ at different $H$ stages. Therefore, to obtain the final simulated sorption isotherm, sorption isotherms at a wide range of pore sizes have to be simulated and then superimposed based on $\psi_i$. This process had been illustrated in detail in [35], and the simulated sorption isotherms could be very close to the experimental ones depending on the choice of $\psi_i$. It is also reported in the literature [52] that a wide pore size distribution can dramatically change the shapes of the sorption isotherms of silica-water systems. The aforementioned process was not done in this work due to the high computation cost of molecular simulations at different pore sizes, especially when several model parameters were also changed. However, this would not affect the main conclusion of this work since the focus here was to evaluate the selected model parameters.

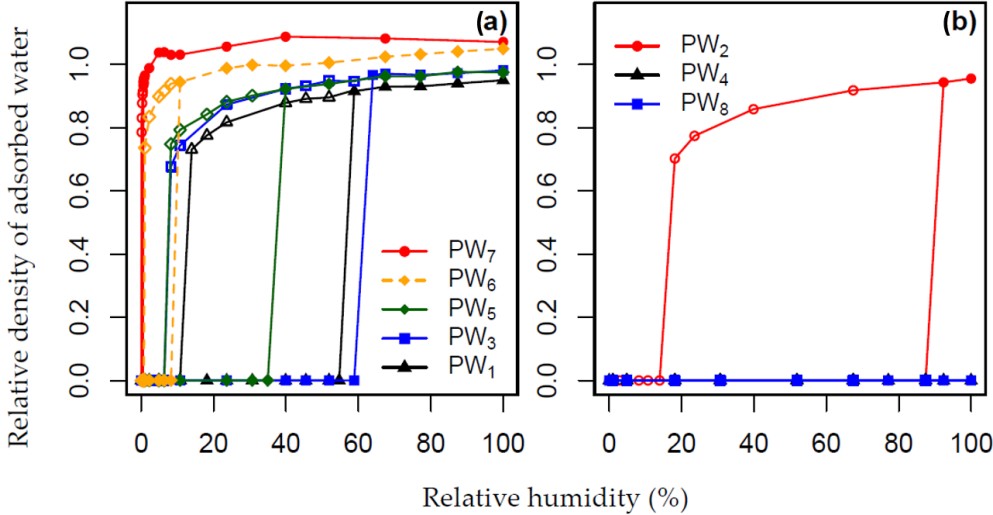

**Figure 4.** Simulated water sorption isotherms from eight types of 0.95 nm cell wall pores at 25 °C with different WWIS. (**a**) $PW_1$, $PW_3$, $PW_5$, $PW_6$ and $PW_7$; (**b**) $PW_2$, $PW_4$ and $PW_8$. Solid symbols, adsorption; open symbols, desorption.

### 3.2. The Validity of the Simulation Model

The developed simulation model had been qualitatively examined using the predicted water sorption hysteresis properties. The magnitude of simulated sorption hysteresis increased with the cell wall pore size, the lignin content of cell walls, and the reduced temperature, which is consistent with experimental observations. Details can be found in [25]. Furthermore, the model had been quantitatively examined by comparing the predicted cell wall pore size distributions at fully saturated states with the experimental ones derived from the solute exclusion method. The predicted distributions were relatively wide with several major peaks evolving in the hygroscopic range but were assessed to be fairly reasonable. Details can be found in [35].

### 3.3. The Effect of Wood-Water Interaction Strength on Simulated Sorption Isotherms

The effect of wood-water interaction strength on simulated sorption isotherms was revealed by comparing the eight sorption isotherms in Figure 4. WWIS was reflected by the LJ parameter $\varepsilon_{sf}$ in Table 1, and larger $\varepsilon_{sf}$ values meant stronger wood-water interactions. Hence, for the simulated pores in Figure 4, WWIS decreased in the following order: $PW_7$ > $PW_6$ > $PW_5$ > $PW_3$ > $PW_1$ > $PW_2$ > $PW_4$ > $PW_8$. It was clear from Figure 4 that WWIS affected both sorption isotherms and hysteresis considerably. Overall, stronger wall-water interaction led to lower $H$ micropore fillings (the abrupt increase on the simulated adsorption isotherm), and narrower hysteresis loops (Figure 4). When the interaction became very strong ($PW_7$), hysteresis almost disappeared, and when the interaction became

very weak ($PW_4$ and $PW_8$), water could not condense in the simulated pores. The sorption isotherm from $PW_3$ (blue lines in Figure 4a), slightly deviated from the general trend since it was supposed to lie between $PW_5$ and $PW_1$ (dark green and black lines in Figure 4a), according to the order of WWIS. This was probably caused by the additional sorption sites on the wall of default $PW_1$. As explained below, the sorption sites could also influence the isotherms and hysteresis significantly. Therefore, the additional sorption sites on $PW_1$ pores could shift the micropore filling of water towards an earlier $H$ stage.

### 3.4. The Effect of Sorption Sites on Simulated Sorption Isotherms

Figures 5 and 6 demonstrate how sorption sites affected the simulated sorption isotherms and hysteresis for $PW_1$ and $PW_2$, respectively. In most cases, micropore filling of water occurred in simulated pores. Stronger sorption site networks brought in by either increased $n_1$, $\varepsilon_H$ or $R_H$ caused lower $H$ micropore fillings and narrower hysteresis loops (Figures 5a–c and 6a–c). The isotherm trends from $PW_2$ (Figure 6) were similar to that of $PW_1$ (Figure 5), but the formers were more sensitive to the sorption sites-related parameter variations, indicating the coupling of the sorption sites and WWIS.

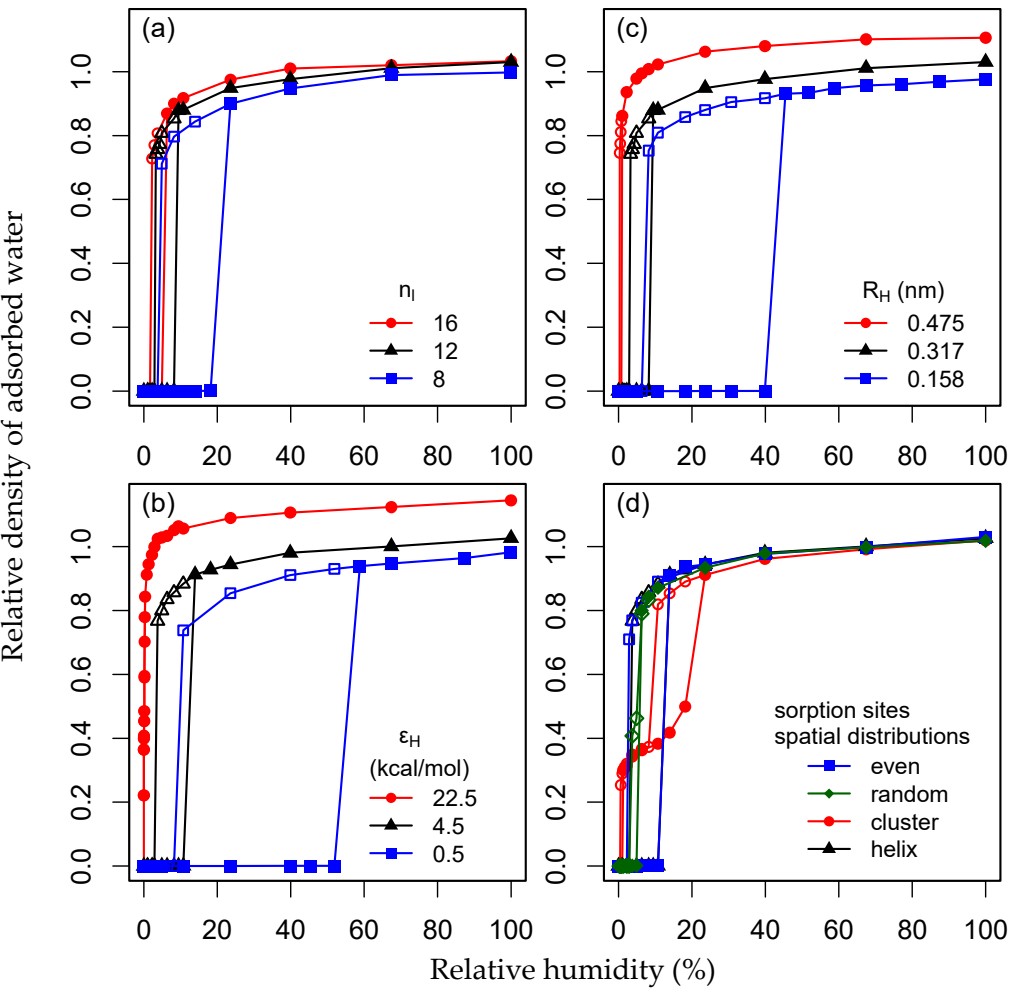

**Figure 5.** Simulated water sorption isotherms of 0.95 nm $PW_1$ pores at 25 °C with different sorption sites (**a**) number, (**b**) strength, (**c**) interaction range and (**d**) spatial distributions. solid symbols, adsorption; open symbols, desorption.

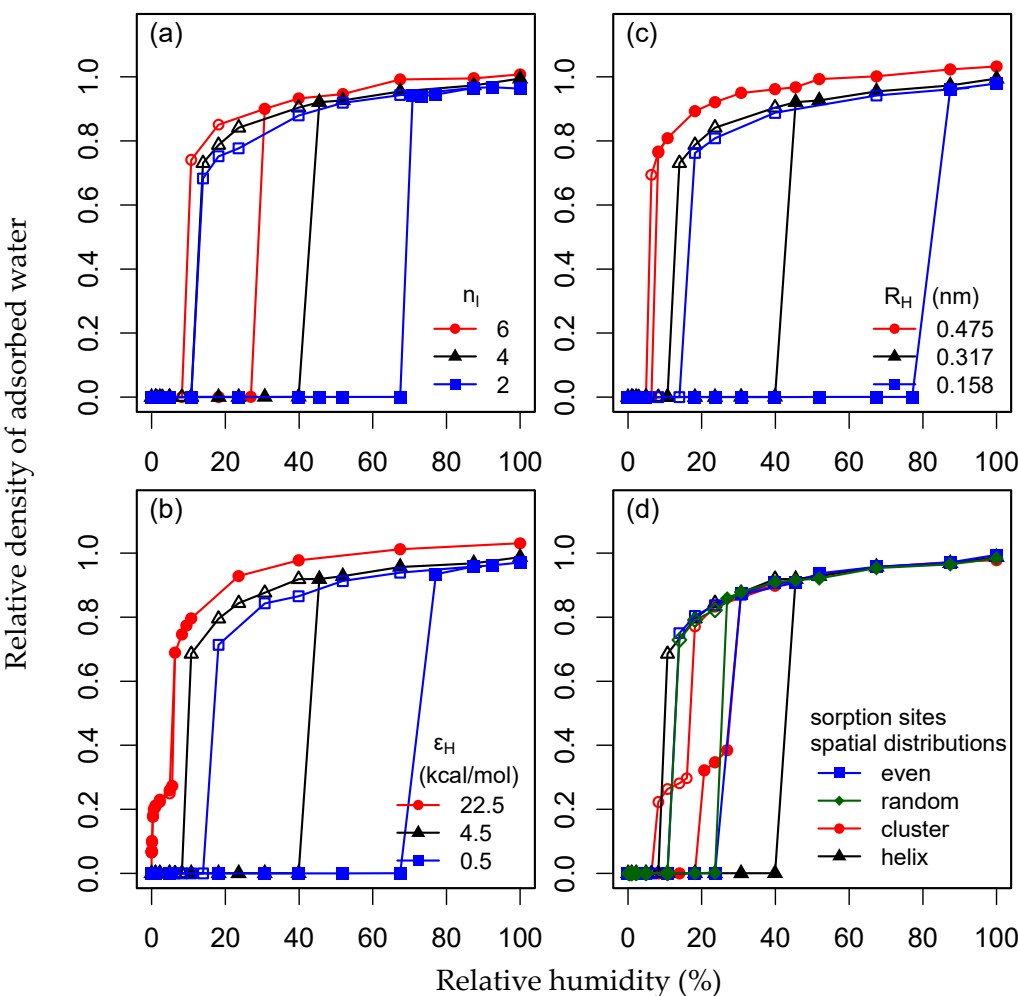

**Figure 6.** Simulated water sorption isotherms of 0.95 nm PW$_2$ pores at 25 °C with different sorption sites (**a**) number, (**b**) strength, (**c**) interaction range and (**d**) spatial distributions. solid symbols, adsorption; open symbols, desorption.

Unlike the aforementioned three sorption site parameters, the site spatial distributions did not significantly affect simulated sorption isotherms and hysteresis. The only exception was for the cases of clustered distributions (Figures 5d and 6d) where a stepping stage on the simulated isotherms was observed. That was caused by a two-stage micropore filling that firstly initiated in the pore region where sorption sites clustered and then other regions.

*3.5. Water Monolayer*

Interestingly, a water monolayer (or film) formed on the surface of the simulated pore at an extremely low *H* of $9.94 \times 10^{-10}\%$ when the WWIS parameter $\varepsilon_{sf}/k_B$ was increased to an extremely high value of 1165.8 K from 58.29 K. Figure 7 gives such a snapshot observed in the simulations. Water molecules were well organized on the surface of the simulated pore walls, compared to the random structure from a normal $\varepsilon_{sf}/k_B$ setting. However, in this specific case, the pore wall-water interaction had been strengthened to an extent that was far beyond the one in a real system, and therefore not likely to exist. Moreover, the simulation here also offers some insights to the small amount of water that cannot desorb from wood even when heated at 105 °C for a long period (defined as water of constitution by Stamm [53]). This type of water is supposed to bound tightly to wood cell wall substances, thus forming a strong wood-water interaction scenario, at least on partial regions of the cell walls. Since the water monolayer could exist at extremely low *H* that is difficult to reproduce experimentally, it is very difficult to remove it in reality.

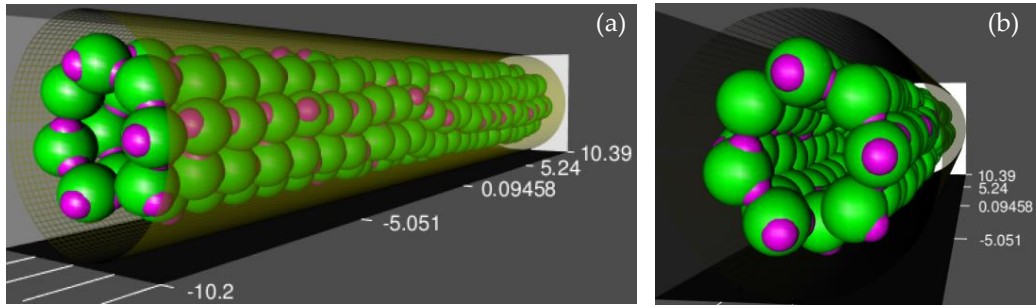

**Figure 7.** Snapshot showing a monolayer formed on a 0.95 nm pore at 25 °C with strengthened wall-water interaction: (**a**) side view and (**b**) front view. Each green ball with two attached magenta balls represents a water molecule.

### 3.6. Water Clusters

Hysteresis disappeared when $\varepsilon_H$ increased to 22.5 kcal/mol (Figures 5b and 6b), and in contrast to abrupt micropore fillings observed in earlier scenarios, a gradual increase of adsorbed water molecules driven by the formation of clusters initiated at the positions of sorption sites was observed. Figure 8 shows snapshots of simulated 0.95 nm pores at a series of given $H$ values: 0.000036%, 0.00050%, 0.16%, 0.35% for $PW_1$ and 0.000036%, 0.60%, 5.57%, 6.34% for $PW_2$. For both pore types, the water clusters were initiated at the sorption sites (indicated in blue balls in Figure 8), and then grew bigger, eventually coalescing as $H$ increased. However, the water clusters only existed at very low $H$ values (<0.35%) for $PW_1$ and slightly higher $H$ values (6.34%) for $PW_2$. Since $PW_2$ has weaker WWIS, the simulation results demonstrated that the less-hydrophilic cell wall pore surfaces benefited the stabilization of the formed water clusters. The sorption process in this special case could be used to examine the situation described by clustering theory mainly developed by Hartley and Kamke and Hartley and Avramidis [5,6]. Accordingly, one water molecule was attached to each sorption site at a low $H$ region (0–30%), and then as $H$ increased, water clusters with an average size of two formed and grew larger. The predicted maximum size of the water cluster was 98 [6]. Apparently, the $H$ region where clusters existed and their maximum size predicted from the clustering theory was not supported by the simulation results here. The set sorption site energy (22.5 kcal/mol) in the simulations was extremely high compared to the normal value of 4.5 kcal/mol, so water sorption driven by cluster formation was not likely to occur in a real wood-water system.

### 3.7. Interaction of Simulation Parameters

The simulated sorption isotherm trends from different parameter settings were similar, and also resembled those from varied cell wall pore sizes in [25]. In summary, the investigated five simulation parameters—pore size, $\varepsilon_{sf}$, $n_H$, $\varepsilon_H$ and $R_H$—largely affected the simulated sorption isotherms and hysteresis. Furthermore, different variation sensibility from $PW_1$ and $PW_2$ indicated interaction among these parameters.

Figure 9 illustrates a typical water sorption isotherm from simulated wood cell wall pores at a given pore size. Two critical points where micropore filling and evaporation occurred, respectively, could be identified (points A and B in Figure 9). The critical $H$s at A and B were denoted as $H_u$ and $H_l$, and then the magnitude of the simulated hysteresis loop could be calculated as $\Delta H = H_u - H_l$. Statistical models were employed to analyze the effects of investigated parameters on $H_u$, $H_l$ and $\Delta H$, and the potential interactions among these parameters based on the pooled simulation data.

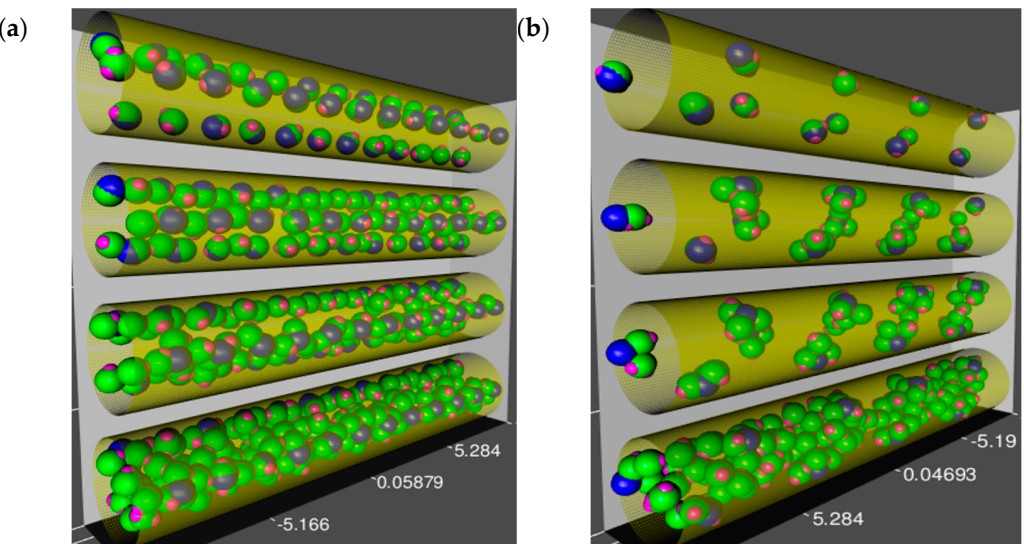

**Figure 8.** Snapshots showing simulated 0.95 nm pores at 25 °C: (**a**) $PW_1$ pores at a series of $H$ from top to bottom: 0.000036%, 0.00050%, 0.16%, 0.35%; (**b**) $PW_2$ pores at a series of $H$ from top to bottom: 0.000036%, 0.60%, 5.57%, 6.34%. Each green ball with attached two magenta balls represent one water molecule, and blues spheres show the position of sorption sites.

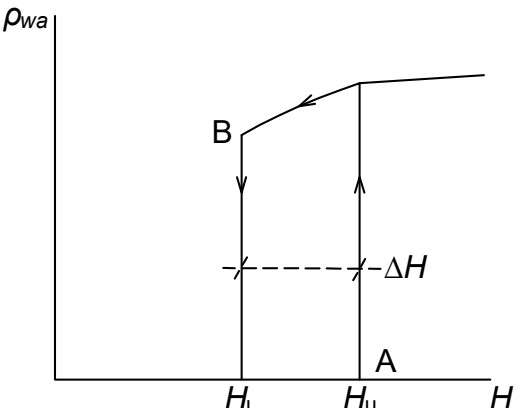

**Figure 9.** Illustration of a typical simulated sorption isotherm in this work.

For the classical Kelvin's equation (Equation (1) in [25], by replacing the surface tension of water $\gamma$ with $\varepsilon_{sf}$ or the combination of $n_H$, $\varepsilon_H$ and $R_H$, two candidate variables, $\frac{\varepsilon_{sf}}{r_p}$ and $\frac{n_H \varepsilon_H R_H}{r_p}$, where $r_p$ (nm) is the radius of the pore, were derived for multiple linear regression analysis. Another variable $\frac{\varepsilon_{sf} n_H \varepsilon_H R_H}{r_p}$ was attempted considering the possible coupling effect from WWIS and sorption sites. Equations (1)–(3) were multiple linear regression models found to fit the simulation data with satisfactory accuracy.

$$lnH_u = \beta_0 + \beta_1 \frac{\varepsilon_{sf}}{r_p} + \beta_2 \frac{\varepsilon_{sf} n_H \varepsilon_H R_H}{r_p} \qquad (1)$$

$$lnH_l = \beta_0 + \beta_1 \frac{\varepsilon_{sf}}{\sqrt{r_p}} + \beta_2 \frac{\varepsilon_{sf} n_H \varepsilon_H R_H}{\sqrt{r_p}} \qquad (2)$$

$$ln\Delta H = \beta_0 + \beta_1 \frac{\varepsilon_{sf}}{r_p} + \beta_2 \frac{n_H \varepsilon_H R_H}{r_p} \qquad (3)$$

Table 3 gives estimated coefficients ($\beta_i$, $i$ = 0 to 2) in Equations (1)–(3) and corresponding adjusted coefficients of determination $R^2$. All the coefficients in Equations (1)–(3) were found statistically significant under the null hypothesis $H_0$: $\beta_i = 0$ with a significance level ($\alpha$) of less than 0.001. The statistical models (1) and (2) indicated that the cell wall pore size, WWIS, and the sorption sites were coupled in affecting the critical properties of sorption isotherms, but the effect from the cell wall pore size was different in determining the critical points on adsorption and desorption isotherms ($H_u$ and $H_l$, respectively). Model (3) indicated the WWIS and the sorption sites were not coupled in determining hysteresis, but both factors were coupled with cell wall pore size.

**Table 3.** Fitted coefficient values in Equations (1)–(3).

|  | $\beta_0$ | $\beta_1$ | $\beta_2$ | Adjusted $R^2$ |
|---|---|---|---|---|
| $H_u$ | 5.2170 | −1.8975 | −0.0051 | 0.9417 |
| $H_l$ | 3.7329 | −2.0409 | −0.0038 | 0.9271 |
| $\Delta H$ | 5.0573 | −2.1227 | −0.0035 | 0.8983 |

Apart from illustrating the interaction of $r_p$, $\varepsilon_{sf}$, $n_H$, $\varepsilon_H$ and $R_H$, the statistical models in this work can be used to predict water sorption isotherms of wood without time-consuming simulations when more accurate input parameters are obtained from experiments.

## 4. Conclusions

The shapes of simulated sorption isotherms in this work were different from the experimental ones, which were caused by focusing on only one cell wall pore size in the simulations due to the computational cost. With proper cell wall pore size distributions further considered, the simulated sorption isotherms would get closer to the experimental ones. Nevertheless, the molecular simulations demonstrated that most investigated parameters including wood-water interaction strength and the number, strength, and interaction range of sorption sites affected both adsorption and desorption isotherms and hysteresis considerably. Water monolayers and clusters were observed on the surface of the simulated cell wall pore at an extremely low humidity when wood-water interaction and sorption site strength was set to extremely high values. However, since the simulation parameters were unrealistically high in these specific cases, water monolayers or clusters were not likely to occur in a real wood-water system. Further multiple linear regressions suggested that the cell wall pore size, wood-water interaction strength, and the sorption sites were coupled in affecting the critical properties of sorption isotherms, but the effect from the cell wall pore size was different in determining the critical points on adsorption and desorption isotherms. Wood-water interaction strength and the sorption sites were not coupled in determining hysteresis, but both parameters were coupled with cell wall pore size.

**Author Contributions:** Conceptualization, J.S. and S.A.; methodology, J.S. and S.A.; software, J.S.; validation, J.S. and S.A.; formal analysis, J.S. and S.A.; investigation, J.S. and S.A.; resources, S.A.; data curation, J.S. and S.A.; writing—original draft preparation, J.S. and S.A.; writing—review and editing, J.S. and S.A.; visualization, J.S. and S.A.; supervision, S.A.; project administration, J.S. and S.A.; funding acquisition, J.S. and S.A. All authors have read and agreed to the published version of the manuscript.

**Funding:** This research was funded by the NSERC Discovery grant RGPIN-2016-04325, and the National Natural Science Foundation of China (Grant No. 32001253), and the Natural Science Foundation of Jiangsu Province (Grant No. BK20200790).

**Institutional Review Board Statement:** Not applicable.

**Informed Consent Statement:** Not applicable.

**Data Availability Statement:** The representative simulation data has been included in the text. Other simulation data will be available upon request.

**Acknowledgments:** The computational resources provided by Compute Canada, and the discussions with Frank Lam from the University of British Columbia and Jiabin Cai from Nanjing Forestry University are acknowledged.

**Conflicts of Interest:** The authors declare no conflict of interest. The funders had no role in the design of the study; in the collection, analyses, or interpretation of data; in the writing of the manuscript, or in the decision to publish the results.

## Abbreviations

| | |
|---|---|
| IUPAC | International Union of Pure and Applied Chemistry; |
| *M* | Moisture content: |
| *H* | Relative humidity: |
| LJ | Lennard-Jones: |
| PW | Pore wall: |
| SPC/E | Extended simple point charge; |
| GCMC | Grand canonical Monte Carlo; |
| WWIS | Wood-water interaction strength |

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
