# Peer review of "Nanopore-Level Wood-Water Interactions—A Molecular Simulation Study"

_forests, doi:10.3390/f12030356_

Round 1

Reviewer 1 Report

I appreciate the effort of the authors in revising the manuscript. However, with regards to my criticism of the fundamental assumptions of the model and absence of model prediction validation, the manuscript is still lacking. 

The claim by the auhors that the two previous papers cited validate the model prediction is not correct. For instance, the fact that a higher lignin content is associated with a larger sorption hysteresis does not validate the assumption that lignin contains larger pores within which water is bound. Also, the claim that model prediction of the pore size distribution in the water-saturated state is validate is incorrect. The experimentally determined pore size distribution is taken directly from solute exclusion measurements without accounting for the exclusion effect of probes with the pore size approaches the size of the probe. This effect has been documented numerous times:

Casassa EF (1967) Equilibrium distribution of flexible polymer chains between a macroscopic solution phase and small voids. Journal of Polymer Science Part B: Polymer Letters 5:773-778 https://doi.org/10.1002/pol.1967.110050907

Dai H, Dubin PL, Andersson T (1998) Permeation of small molecules in aqueous size-exclusion chromatography vis-à-vis models for separation. Analytical Chemistry 70:1576-1580 https://doi.org/10.1021/ac970968m

Day JC, Alince B, Robertson AA (1979) The characterization of pore systems by macromolecular penetration. Cellulose Chemistry and Technology 13:317-326

Dubin PL, Edwards SL, Mehta MS, Tomalia D (1993) Quantitation of non-ideal behavior in protein size-exclusion chromatography. Journal of Chromatography A 635:51-60 https://doi.org/10.1016/0021-9673(93)83113-7

The model predictions do not match those of the solution exclusion data anyway, but it is incorrect to state that the model has been validated in terms of the fundamental assumptions concerning sorption mechanism, pore sizes or any other physical parameters of the model.

Furthermore, it is incorrect to state that the model is closer to the real system, just because it includes certain claims about the physical nature of water sorption in wood. Being different from or having more parameters than the Hailwood-Horrobin model, BET model or any other sorption isotherm model is not the same as being more physical realistic. It does not matter that the model makes all sorts of predictions about pore size distributions or other parameters if these predictions are not evaluated against independently determined values from experiments.

L55-58: “Comparing to the purely theoretical and experimental approach, the molecular simulation approach has the advantage of conducting systematic and quantitative investigations at the nanoscale using a model that is close to the real system.” The authors have not shown that their approach differs from e.g. the Hailwood-Horrobin model that treats water sorption in polymeric materials by solution thermodynamics. Not that this model is necessarily true in anyway, but the claim that the kind of molecular simulation model used in this study is close to the real system is not supported in this or previously published papers.

Reviewer 2 Report

„Nanopore-Level Wood-water Interactions—A Molecular Simulation Study” is a well-written and very interesting paper focused on wood-water interactions studied using a cell wall model. Although these relationships seem to be already well recognised, there are still a lot of questions about the mechanisms driving the process of binding and releasing water molecules by the wood cell wall. Although this is research based on simulations, it helps to understand the phenomena that occur in the cell wall pore surface when it binds water molecules. Some comments and suggestions:

  • It is hard for a reader to follow all references in the text (references to other methods, details, etc.) to fully understand the article. Would it be possible to describe briefly the most important issues/conclusions in such places instead of giving only a reference, please?
  • Lines 172-173 –the reference format is incorrect (a consecutive number instead of a year of publication).
  • Snapshots showing water in the simulated pores are very informative.
  • I look forward to reading your future work on this topic.

Author Response

This manuscript is a resubmission of an earlier submission. The following is a list of the peer review reports and author responses from that submission.

Round 1

Reviewer 1 Report

I have previously reviewed this manuscript for another wood journal, where it was rejected because of insufficient quality. Unfortunately, the manuscript submitted to Forests has not considered the previously obtained reviewer comments. The following is therefore in large parts a re-iteration of previously provided critique.

The manuscript concerns the simulation of water sorption in wood based on a previously developed simplified molecular simulation model. The hallmark of a good theoretical model is that it can predict experimentally observed parameters (e.g. moisture content as function of RH) based on other, independently experimentally determined, parameters. Although the manuscript covers in detail how the different parameter values of the model affect the model predictions, there is no discussion of the validity of the selected parameter values or indeed of the simplified model itself in describing water molecules interacting with wood. The authors do state that “the molecular simulation approach has the advantage of conducting systematic and quantitative investigations at the nanoscale using a model that is close to the real system.” (Lines 56-59), but there is no discussion of whether the utilised model fulfils the criterion of being “close to the real system”.  In fact, the model is not in accordance with the current understanding of the wood cell wall as it models this as a porous material with cylindrical pores of specific sizes, where sorption sites are located on the pore walls. When the moisture content increases, the pore size and distribution is changed in order to obtain the sigmoid shape observed for sorption isotherms of wood. The mechanisms for water sorption used in the model are “water layering” and “clustering” which the authors describe as “two established sorption mechanisms at low to medium relative humidity” (Lines 34-35), however, none of these concepts are established in the research field as there is still no direct evidence or consensus about exactly how water molecules are associated with the cell wall. In the end, the work is purely theoretical in nature and does not advance our understanding of water in wood.

Reviewer 2 Report

A very interesting piece of work which models sorption behaviour in a system with rigid pores and describes the sorption isotherm. The behaviour described is not surprisingly very close to that observed with systems containing rigid pores, such as sandstone, charcoal, zeolites, etc. The model does not successfully replicate the behaviour observed with wood. The authors do not discuss this and do not mention this in their conclusions. Wood exhibits swelling behaviour during sorption, which is extremely important when considering sorption. This needs to be discussed and part of the conclusion has to be that the model does not replicate the sorption behaviour of wood. They also need to discuss the sorption behaviour of rigid systems with some examples and discuss how their model replicates behaviour in such cases. A revision on the MS is therefore necessary to take the above into account.